

# Effects of quarantine on Physical Activity prevalence in Italian Adults: a pilot study

Mario Mauro[1,*], Stefania Toselli[2,*], Silvia Bonazzi[2], Alessia Grigoletto[2], Stefania Cataldi[3], Gianpiero Greco[3] and Pasqualino Maietta Latessa[1]

[1] Department of Life Quality Studies, University of Bologna, Bologna, Italy
[2] Department of Biomedical and Neuromotor Sciences, University of Bologna, Bologna, Italy
[3] Basic Medical Sciences, Neurosciences and Sense Organs, University of Bari, Bari, Italy
* These authors contributed equally to this work.

Corresponding authors
Stefania Cataldi,
stefania.cataldi@uniba.it
Gianpiero Greco,
gianpiero.greco@uniba.it

## ABSTRACT

**Background:** COVID-19 is a respiratory disease that caused a global pandemic status in March 2020. Due to its fast diffusion, many governments adopted forced solutions including social restrictions, which could negatively affect citizens' habits as physical activity. Our study aimed to investigate how and why the physical activity prevalence varied from the period before the quarantine up to the period after it, and understand what citizens thought of physical inactivity COVID-19 related to and whether they were satisfied with physical activity promotion during the lockdown.

**Methods:** A new questionnaire was created and administered online. A sample of 749 interviews (female = 552 (73.7%), male = 197 (26.3%)) was collected and analysed.

**Results:** The prevalence of people who were older than 50 years reduced both during and after the lockdown ($P < 0.05$) and the most common reason for which they have quitted physical activity practice was related to psychological problems (lockdown = 64.57%; post-lockdown = 62.17%). In addition, youngers seemed to be more sensitive than elders to unhealthy consequences generated by forced isolation ($P < 0.05$), and they believed that children/adolescents and older adults practised an insufficient amount of physical activity and/or sport, which could negatively impact public health.

**Conclusions:** Although many strategies were implemented during the lockdown to promote regular physical activity practice, several results suggested that quarantine negatively affected citizens' habits. The future government should focus on adequate measures to improve health behaviours.

## INTRODUCTION

Coronavirus 2019 (COVID-19) is a respiratory disease caused by severe acute respiratory syndrome coronavirus 2 (SARS-CoV-2) that was first noted as a case of unexplained pneumonia in December 2019 (*Guo et al., 2020*). Since COVID-19 showed fast human-to-human transmission through aerosols, droplets, and close contact, the World Health Organization (WHO) declared a global pandemic on March 11 (*World Health Organization, 2022*). The earliest appearance of COVID-19 in Italy was on January 30 and

after 2 weeks it spread progressively over the national territory (*La Maestra, Abbondandolo & De Flora, 2020*). As a method to control and reduce individual interaction, many countries used a combination of several non-pharmaceutical interventions such as the isolation of infected people, the tracking of exposed people, travel restrictions, lockdown, wearing masks and frequent sanitization (*Goenka & Liu, 2022*). Italian government adopted strategies such as quarantine and social restrictions, which had consequences on social and economic aspects.

However, authorities and policymakers are supposed to consider how these preventive measures impacted citizens' lives and public health. Some researchers found that social restrictions negatively affected economic and employment status, reduced education levels decreased general well-being, and physical and mental health of several social groups (*Diedhiou, Yilmaz & Yilmaz, 2021*; *Osterrieder et al., 2021*). Many studies reported that quarantine and isolation measures were associated with an increased risk of experiencing mental health burdens, especially in people who have reported previously disorders (*Regehr et al., 2021*; *Wang et al., 2021*). In addition, a review evidenced that longer quarantine duration, infection fears, frustration, boredom, inadequate supplies, inadequate information, financial loss, and stigma acted as stressors, which negatively affected psychological aspects (*Brooks et al., 2020*).

As a strategy of prevention, promoting healthy habits has elicited interest. Of these, the regular practice of physical activities (PA) plays a key role in preventing the onset of cardiovascular, metabolic, cancer, and neuromuscular diseases, such as viral infections, and it could reduce the risk of hospitalization, morbidity, and mortality (*Liu et al., 2019*; *Damiot et al., 2020*; *Lippi, Henry & Sanchis-Gomar, 2020*; *Silva Filho et al., 2020*; *Després, 2021*). Also, the daily practice of PA can delay senescent muscle atrophy, and fragility, can reduce the risk of fall, and generates mental health benefits in older adults (*Campa, Silva & Toselli, 2018*; *Cruz-Jentoft et al., 2019*; *Callow et al., 2020*; *Campa et al., 2021*). To promote and facilitate the PA practice, the WHO recommended easy guidelines for different populations (*World Health Organization, 2020*). Despite this, many studies showed that social restrictions negatively impacted people's habits and made them less active (*Lesser & Nienhuis, 2020*; *Gjaka et al., 2021*; *Zaccagni, Toselli & Barbieri, 2021*; *Puccinelli et al., 2021*; *Mauro et al., 2022*; *Dallolio et al., 2022*).

Although several researchers investigated the PA parameters during the lockdown, to the best of our knowledge no authors have analysed the prevalence of the PA practice in Italian citizens over the pandemic. Our main hypothesis saw a reduced prevalence after the quarantine, expecting it negatively impacted the citizens' habits. Also, more information on what citizens thought of socio-political government strategies and whether these could have affected people's lives is needed. Therefore, this study aimed (1) to develop a questionnaire that analyses the effect of quarantine on PA and/or sports by comparing the pre, during, and post-lockdown prevalence among gender and age classes, and (2) to research opinions regarding the link between the PA and lockdown, and (3) to investigate whether Italian citizens retained that government strategies could have caused negative consequences for public health.

**Table 1 Socio-cultural characteristics of people interviewed.**

|  | Category | Freq | (%) | n |
|---|---|---|---|---|
| Gender | Male | 197 | 26.3 | 749 |
|  | Female | 552 | 73.7 |  |
| Education | Primary School | 4 | 0.53 | 749 |
|  | Secondary School | 75 | 10.01 |  |
|  | High School | 369 | 49.27 |  |
|  | Degree | 163 | 21.76 |  |
|  | Master's degree | 118 | 15.75 |  |
|  | PhD | 14 | 1.87 |  |
|  | Other | 6 | 0.80 |  |
| Occupation | Employed | 414 | 55.27 | 749 |
|  | Freelance | 115 | 15.35 |  |
|  | Retired | 75 | 10.01 |  |
|  | Student | 73 | 9.75 |  |
|  | Unemployed | 29 | 3.87 |  |
|  | Homemade | 15 | 2 |  |
|  | Other | 28 | 3.74 |  |
|  | Mean (±SD) | Min | Max | n |
| Age (year) | 44.14 (14.55) | 18 | 84 | 749 |
| Category | Mean (±SD) | Freq | (%) |  |
| 18–39 | 28.28 (5.46) | 287 | 38.32 |  |
| 40–49 | 45.12 (2.63) | 157 | 20.96 |  |
| 50–64 | 55.96 (3.85) | 245 | 32.71 |  |
| >64 | 69.15 (4.21) | 60 | 8.01 |  |

**Note:**
SD, standard deviation; n, number of observations; Freq, frequencies.

# MATERIALS AND METHODS

## Participants and study design

The current study is a retrospective observational design, which used a questionnaire to record and gather data. The surveys were shared on the Web from 2[nd] December 2020 up to 10[th] January 2021 and involved 749 participants (female = 552 (73.7%), male = 197 (26.3%), Table 1). During this period, the Italian Government declared the end of many social restrictions, allowing citizens to practice all types of physical activity in outdoor spaces, and few physical activities in indoor spaces, in respect of every emergency measure (wearing a mask, social distancing of almost one meter, body-temperature measurement, and hand sanitation).

The questionnaire selection criteria included adult people who were at least 18 years old, who lived in Italy during the COVID-19 pandemic, and who spoke the Italian language. No country or region limitations were explained. Written informed consent was provided by the participants before the study began. The study was approved by the Bioethics Committee of the University of Bologna (Approval code: 224254, October 9, 2020).

## Questionnaire

We created the questionnaire with Google Forms and shared it on one global social network (Facebook®; Meta Platforms, Inc, Cambridge, MA, USA) and WhatsApp® (Meta Platforms, Inc, Cambridge, MA, USA). All participants were informed and gave us privacy consent to treat their data. They could fill out the survey with no Google sign-in request. They could manually enter all general information or allow social networks to report them. The questionnaire was self-administered in the Italian language. Each completed survey was saved on a Google database, and we gathered all data as an Excel spreadsheet (Microsoft Office®; Microsoft Corporation, Redmond, WA, USA).

The questionnaire was of two parts: the first part included general and demographic information such as age (in years), sexes (male or female), educational status (primary school, secondary school, high school, degree, master's degree, PhD, or other), occupational status (student, employed, freelance, unemployed, homemade, retired, other), information on PA prevalence over the quarantine (example: "Did you practice PA and/or sports before the COVID-19 quarantine?"), and information about motivations that negatively or positively affected citizens' PA habits (example: "If you did not practice any PA during the lockdown, why?"); the last part included 13 questions that investigated whether the sample believed physical inactivity could negatively impact people's health and whether the sample thought that socio-political strategies affected the citizens' habits.

Several types of data were collected and transformed before the analysis. Questions on PA prevalence and gender were recorded as binary data (yes or no, male or female) and transformed in Bernoulli data (0 or 1) where 0 was not and female, and 1 was yes and male. The citizens' age was collected as years and successively grouped into four categories (a) 18–39 years (young adults), (b) 40–49 years (early middle-aged adults), (c) 50–64 years (late middle-aged adults), and (d) ≥65 years (old adults). A previous study reported four age intervals stratified into 12 age subsets (*Horng, Lee & Chen, 2001*); however, we divided the middle-aged category into two groups and we settled on 65 years as the cut-off point for the elder category (Italian retired age). Information that investigated why citizens did not practise PA were registered as nominal data with five options for the period during and after the lockdown respectively: "COVID-19 problems", "Environmental problems", "Health problems", "Psycho-social problems", "Technological problems"; "COVID-19 problems", "Environmental problems", "Health problems", "Psychological problems", "Socio-economics problems". Differently, the reasons for which people did PA and/or sport during lockdown were: "General well-being", "Physical well-being", "Psychological well-being", "Physical and General well-being", "Psychological and General well-being", and "Psychological and Physical well-being". Finally, the 13 items were recorded as ordinal data such as "disagree, partially disagree, partially agree and agree", and then transformed into discrete items (Likert-type scale) from 1 (disagree) to 4 (agree).

## Statistical analysis

We reported the frequencies of occurrence as the number of observations for categorical and binary data and mean ± standard deviation (SD) and the minimum and maximum observed values for numerical (continuous and discrete) data. The prevalence of PA was

analysed among sexes and age groups: the McNemar's test ($\chi^2$) was assessed with the exact McNemar significance probability (*P*) (*Brunner & Giannini, 2011*). In addition, the Risk Ratio (RR) was calculated as the previous study suggested (*Ranganathan, Aggarwal & Pramesh, 2015*).

The reliability of the questionnaire items was tested through homogeneity and internal consistency. To identify the dimensions of our questionnaire, the exploratory factor analysis (EFA) of tetrachoric correlations was assessed (*Schreiber, 2021*). To measure the sampling adequacy, the Kaiser–Meyer–Olkin (KMO) value was calculated; values >0.80 were considered meritorious. To test the null hypothesis that variables were not intercorrelated, Bartlett's test of sphericity was performed, and the determinant of matrix correlation value, $\chi^2$, degree of freedom (df), and the *p*-value (*P*) were reported. The choice of the number of factors was based on the eigenvalues, and we used the unweighted least-squares method and the Kaiser rule to extract only factors with an eigenvalue ≥1. Finally, the orthogonal Varimax rotation was used and the related $\chi^2$ and *p*-values (*P*) were settled. Items with a loading value <0.40 were dropped, and the final model included only items with loadings of ≥0.40 on their specific factors. In addition, to give a measure of the common variance proportion of the variable not associated with the factors, the Uniqueness (1−commonality) was calculated (*Lawley & Maxwell, 1962*).

Then, to provide the internal consistency of our questionnaire, we calculated Cronbach's alpha on all items and each factor, respectively. We considered the alpha value acceptable ranging from 0.70 to 0.95 (*Tavakol & Dennick, 2011*), and we reported the average interitem correlation and alpha values ($\alpha$). Finally, the test-retest reliability was assessed through Spearman's rank correlation coefficient ($\rho$), where a value equal to 1 was considered perfect reliability, a value between 1 and ≥0.90 was considered excellent, a value ≥0.80 and ≤0.90 was considered good, a value ≥0.70 and ≤0.80 was considered acceptable, a value ≥0.60 and ≤0.70 was considered questionable, and all values <0.60 was considered poor reliable (*Shou, Sellbom & Chen, 2022*).

The ANOVA methods were used to compare the mean differences among age groups, sexes, and age-sexes interactions. The Snedecor-Fisher test value (F) and *p*-value (*P*) were reported.

We selected *a priori* hypothesis test significance levels as equal to 0.05 ($\alpha$). In addition, we used G*Power 3.1.9.7 for Windows 10 (Heinrich-Heine-Universitat Düsseldorf, Universitätsstraße 1, 40225 Düsseldorf, Germany) to compute an *a priori* sample size analysis. which equalled to 731 with the following parameters: type I error = 0.05, Effect Size = 0.15, Statistical Power = 0.80 and number of groups = 8 (gender*Age). All statistical analyses were performed by STATA® software (version 17, StataCorp., College Station, TX, USA, StataCorp LP).

# RESULTS

## Sample characteristics

We collected 751 interviews. Two of these were excluded for missing values and we finally analysed 749 interviews.
**Table 2 Differences in PA frequencies before, during and post lockdown, among sexes and age groups.**

| | | PA pre lockdown | | PA lockdown | | PA post lockdown | | Δ PA (pre-lock) | | | Δ PA (pre-post) | | | Δ PA (lock-post) | | |
|---|---|---|---|---|---|---|---|---|---|---|---|---|---|---|---|---|
| | | No | Yes | No | Yes | No | Yes | $\chi^2$ | P | RR | $\chi^2$ | P | RR | $\chi^2$ | P | RR |
| Gender | F | 176 | 376 | 195 | 357 | 240 | 312 | 2.27 | 0.15 | 0.90 | 34.13 | <0.001* | 0.73 | 16.46 | <0.001* | 0.81 |
| | M | 49 | 148 | 59 | 138 | 64 | 133 | 2.50 | 0.15 | 0.83 | 8.33 | <0.01* | 0.76 | 0.64 | 0.52 | 0.92 |
| Age | 18–39 | 87 | 196 | 77 | 210 | 100 | 187 | 1.47 | 0.27 | 1.13 | 3.45 | 0.09 | 0.91 | 8.67 | <0.01* | 0.48 |
| | 40–49 | 49 | 108 | 55 | 102 | 71 | 86 | 0.82 | 0.45 | 0.89 | 14.24 | <0.001* | 0.69 | 6.40 | 0.01* | 0.77 |
| | 50–64 | 71 | 174 | 95 | 150 | 104 | 141 | 8.23 | <0.01* | 0.75 | 20.55 | <0.001* | 0.68 | 1.65 | 0.25 | 0.91 |
| | >64 | 18 | 42 | 27 | 33 | 29 | 31 | 4.76 | <0.05* | 0.67 | 11.00 | 0.001* | 0.62 | 0.33 | 0.77 | 0.93 |
| Total | | 225 | 524 | 254 | 495 | 304 | 445 | 4.23 | <0.05* | 0.88 | 42.46 | <0.001* | 0.74 | 15.43 | <0.001* | 0.83 |

Notes:
* Significant level.
pre, before the lockdown; lockdown, during the lockdown; post, after the lockdown; $\chi^2$, McNemar $\chi^2$; RR, Risk Ratio; P, Exact McNemar significance probability.

Table 1 shows participants' characteristics of gender type, educational level, occupational kind, and age. Most of the sample was female and younger than 50 years old, and approximately half participants had high school graduation. Just 3.87% of the sample was unemployed.

## Physical activity prevalence

Table 2 shows the prevalence of PA before, during, and after the forced lockdown, among age groups and sexes. Generally, a decreasing prevalence of PA trend was observed with significant differences between the period before lockdown and periods during and after. When stratified for gender, no significant differences emerged between the periods before and during the lockdown in both sexes, while the number of women and men who practised PA before lockdown decreased after quarantine drastically. As regards the age classes, a negative trend was observed for the elder categories (≥40 years), which decreased both during and after the lockdown. The number of younger individuals (18–39) who remained active during lockdown did not decrease significantly, but a greater reduction was observed after the quarantine.

In sexes analysis, female people showed a negative trend with lower prevalence post the lockdown, whereas male subjects did not report significant differences.

In age comparisons, a significant reduction of people who practised PA before the lockdown was observed post the lockdown in subjects who were 40–64. Also, a lower prevalence was recorded post-lockdown than during lockdown in the youngest group.

Successively, the motivations for which individuals chose to practice (or not) PA and/or sport during lockdown were investigated. Generally, most of the sample who did not practice PA and/or sport experienced Psycho-social problems (64.57%), while 16.54% reported Environmental problems. Only 10 individuals (3.94%) had problems with technology. When gender differences were analysed, most of both male and female samples were inactive due to Psycho-social problems (M = 35, 59.32%; F = 129, 66.15%); also, the percentage of women and men who did not practice PA and/or sport for Health problems was similar, 8.72% and 8.47% respectively. In addition, Fig. 1A shows the percentage of responses divided by gender and age. Differently, 74.95% of the sample who

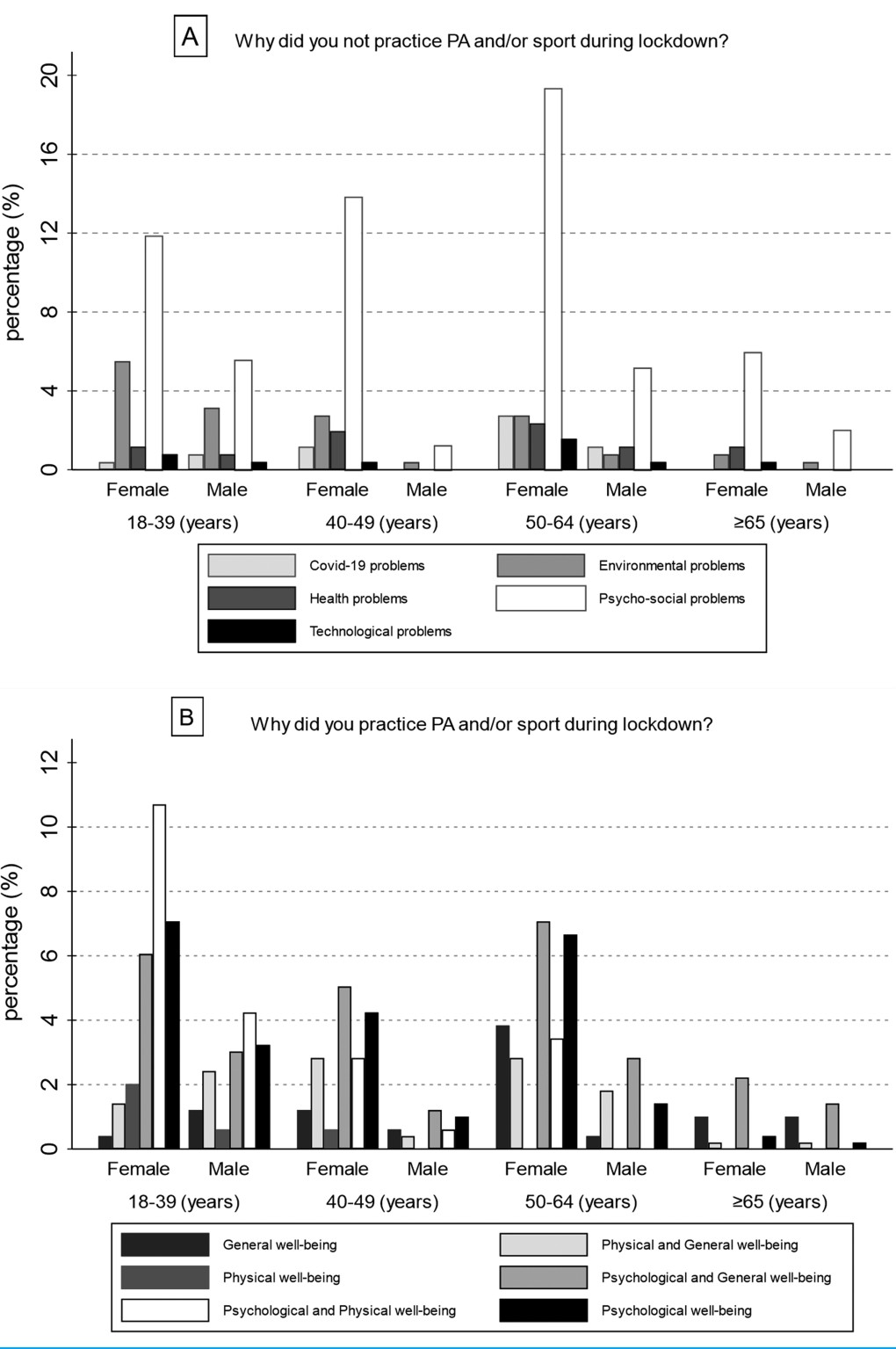

**Figure 1 Motivations which led Italian citizens to do not practice (A) or practice (B) physical activity and/or sport during lockdown.**

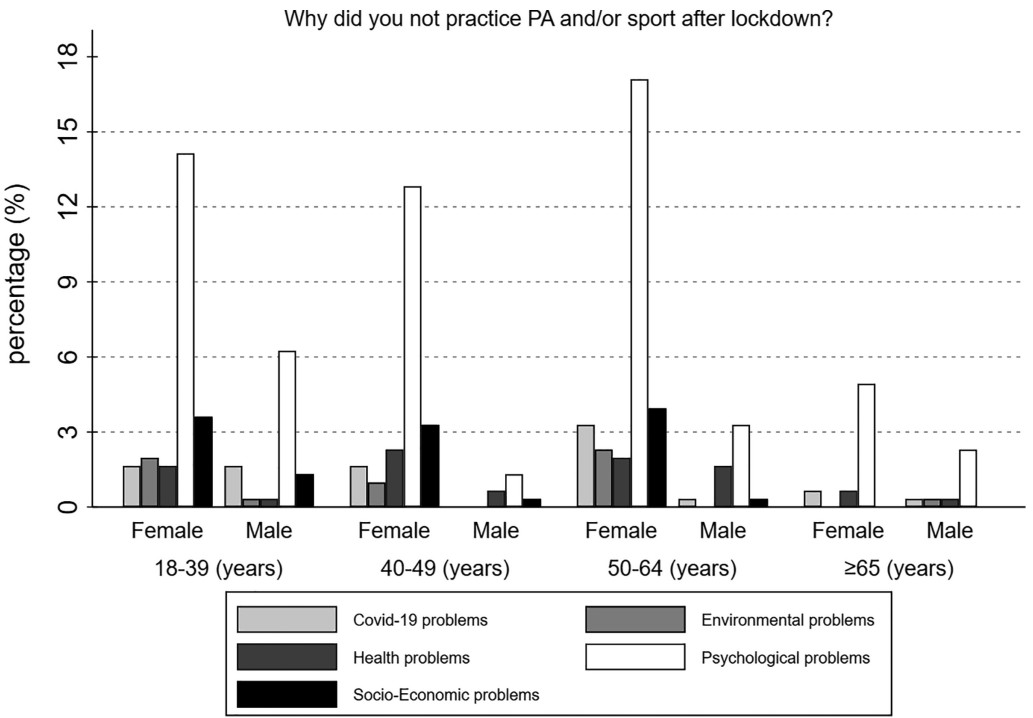

**Figure 2** Motivations which led Italian citizens to do not practice PA and/or sport after lockdown.

were active during the lockdown selected answers with psychological benefits, while just 3.23% practised PA and/or sport for physical well-being alone. This prevalence in psychological responses remained same in gender analysis too, despite was more prevalent in women (F = 276, 77.31%; M = 95, 68.84%). Figure 1B shows the percentage of responses divided by gender and age.

In addition, the reasons why individuals chose to not practice PA and/or sport after the lockdown was investigated. Generally, 189 people (62.17%) did not practice PA and/or sport due to Psychological problems, while 12.83% experienced Socio-Economic problems. Just 18 individuals (5.92%) were influenced by Environmental problems. When gender differences were analysed, 62.08% of women and 62.5% of men experienced Psychological problems, while the minor problems for both females and males were Environmental related (F = 16, 6.67%; M = 2, 3.12%). Finally, Fig. 2 shows the percentage of responses divided by gender and age.

## Questionnaire characteristics

We included 11 items in the model (Table 3). In the beginning, 18 questions were considered, of which five were excluded because data was not on the Likert scale and two were excluded due to poor loading values. Each item was labelled as PAA#, which indicates Physical Activity Aspects and the number (order) of the question. The questions were the following:

1. PAA2: "Do you think the lack of practice of physical activity and/or sport during lockdown has negatively affected your health?"

2. PAA3: "Do you think the lack of practice of physical activity and/or sport in older adults during lockdown has negatively affected their health?"
3. PAA5: "Do you think the lack of practice of physical activity and/or sport during lockdown could cause problems for the national health system and public health?"
4. PAA6: "Do you think the lack of practice of physical activity and/or sport could have affected emotional/psychological wellbeing during the lockdown?"
5. PAA7: "Do you think the lack of practice of physical activity and/or sport could have affected emotional/psychological growth in children/adolescents during the lockdown?"
6. PAA8: "Do you think children/adolescents have practised enough amount of physical activity and/or sport during the lockdown?"
7. PAA9: "Do you think the lack of practice of physical activity and/or sport could have affected emotional/psychological status in older adults (>65 years old) during the lockdown?"
8. PAA10: "Do you think the Italian government adequately considered the practice of physical activity and/or sport during the lockdown?"
9. PAA11: "Do you think the usage of tech devices to practice physical activity and/or sport during lockdown could have affected social interaction negatively?"
10. PAA12: "Do you think sports federations and societies adequately promoted the practice of physical activity and/or sport during the lockdown?"
11. PAA13: "Do you think public and private schools adequately promoted the practice of physical activity and/or sport during the lockdown?"

Firstly, we analysed the sample adequacy (KMO = 0.81), the correlation matrix determinant (Det = 0.09), and the Bartlett's test of sphericity ($\chi^2$ = 1,743.66; dfs = 55; $P < 0.001$). Figure 3 shows the scree plot of eigenvalues with the Kaiser rule. Two factors met our criteria and the Varimax rotation reported a LR test of significant results ($\chi^2$ = 1,746; $P < 0.001$; $n$ = 749). Table 3 shows that factor 1 (how participants thought physical inactivity impacted people's health) included seven items and factor 2 (what participants thought about socio-political strategies on PA habits) included four items. In addition, we assessed Cronbach's alpha on 11 items. The last row of Table 3 shows all Cronbach's alpha values. We used the mean test scale on standardized items, deleting missing values from the analysis. The average interitem correlation on 749 observations was 0.097 and the scale reliability coefficient ($\alpha$) was 0.724. In addition, we reported each consistency factor data: factor one $\alpha$ was equal to 0.779, whereas factor two $\alpha$ equalled 0.603.

After assessing EFA computations, test-retest reliability was assessed in a sample of 149 citizens (Table 4). The retest period was of 1 month. The lowest value of $\rho$ was 0.738 (acceptable), while the highest value was equal to 0.914 (excellent).

**Table 3 Rotated factor loadings (pattern matrix) and Cronbach's alpha values.**

| Items (11) | Factor 1 | Factor 2 | |
| --- | --- | --- | --- |
| | 7 | 4 | Uniqueness |
| PAA6 | 0.771 | −0.062 | 0.4 |
| PAA3 | 0.731 | −0.01 | 0.464 |
| PAA5 | 0.722 | −0.012 | 0.478 |
| PAA2 | 0.71 | −0.111 | 0.483 |
| PAA9 | 0.634 | −0.022 | 0.598 |
| PAA7 | 0.627 | 0.129 | 0.59 |
| PAA11 | 0.415 | 0.019 | 0.827 |
| PAA13 | −0.045 | 0.766 | 0.411 |
| PAA12 | 0.119 | 0.676 | 0.529 |
| PAA8 | −0.067 | 0.623 | 0.607 |
| PAA10 | −0.273 | 0.623 | 0.537 |
| $\alpha = 0.724$ | $\alpha = 0.779$ | $\alpha = 603$ | |

**Note:**
Loadings ≥ 0.4 are shaded.

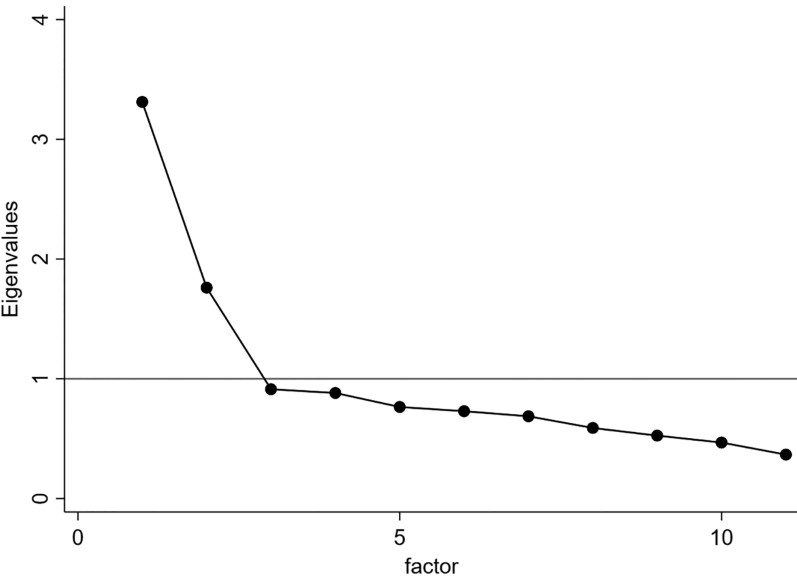

**Figure 3 Scree-plot.** The grey horizontal line (Y = 1) indicates the value under which factors were excluded.

## Physical activity aspects

Table 5 shows the ANOVA outcomes. As regards factor 1, both female and younger subjects exhibited higher value than men, which means they believed that physical inactivity due to COVID-19 social restrictions impacted the psychological status of adolescents and adults and will negatively affect global health. About factor 2, male and elder people reported significantly higher values than women and youths on items 8 and 10, which indicates that younger women thought of Italian government adopted

**Table 4  Test-retest correlation coefficients in 149 citizens.**

| Variable | ρ | P |
|---|---|---|
| PAA2 | 0.738 | <0.001* |
| PAA3 | 0.815 | <0.001* |
| PAA5 | 0.914 | <0.001* |
| PAA6 | 0.778 | <0.001* |
| PAA7 | 0.814 | <0.001* |
| PAA8 | 0.825 | <0.001* |
| PAA9 | 0.899 | <0.001* |
| PAA10 | 0.907 | <0.001* |
| PAA11 | 0.747 | <0.001* |
| PAA12 | 0.826 | <0.001* |
| PAA13 | 0.756 | <0.001* |

Notes:
* Statistically significant.
ρ, Spearman's ρ correlation coefficient; P, p-value; n, number of observations.

**Table 5  ANOVA outcomes of factor 1 and 2 aspects related to the PA practice, by sexes and age groups.**

| Factor 1 | 18–39 years (287) | | 40–49 years (157) | | 50–64 years (245) | | >65 years (60) | | Sexes | | Age | | Sexes × Age | | Range |
|---|---|---|---|---|---|---|---|---|---|---|---|---|---|---|---|
| | M (100) | F (187) | M (23) | F (134) | M (54) | F (191) | M (20) | F (40) | F | P | F | P | F | P | |
| | Mean (±SD) | | Mean (±SD) | | Mean (±SD) | | Mean (±SD) | | | | | | | | |
| PAA6 | 3.44 (0.66) | 3.60 (0.56) | 3.43 (0.59) | 3.57 (0.51) | 3.26 (0.68) | 3.34 (0.57) | 2.90 (0.72) | 3.32 (0.65) | 8.61 | <0.01* | 12.01 | <0.01* | 7.17 | <0.01* | 1–4 |
| PAA3 | 3.31 (0.69) | 3.25 (0.78) | 3.26 (0.45) | 3.33 (0.66) | 3.11 (0.72) | 3.21 (0.69) | 2.90 (0.79) | 3.10 (0.81) | 0.41 | 0.52 | 2.87 | <0.05* | 1.59 | 0.13 | 1–4 |
| PAA5 | 2.85 (0.69) | 3.05 (0.68) | 2.83 (0.78) | 3.03 (0.76) | 2.81 (0.70) | 2.96 (0.67) | 2.65 (0.81) | 2.80 (0.72) | 9.64 | <0.01* | 2.14 | 0.09 | 2.30 | <0.05* | 1–4 |
| PAA2 | 3.18 (0.64) | 3.25 (0.72) | 3.30 (0.70) | 3.25 (0.70) | 2.98 (0.71) | 3.09 (0.67) | 2.60 (0.60) | 3.15 (0.70) | 3.32 | 0.07 | 4.95 | <0.05* | 3.62 | <0.05* | 1–4 |
| PAA9 | 2.85 (0.80) | 3.09 (0.79) | 2.87 (0.87) | 3.07 (0.82) | 2.87 (0.78) | 2.99 (0.71) | 2.80 (0.83) | 3.03 (0.70) | 9.34 | <0.01* | 0.40 | 0.75 | 2.63 | <0.05* | 1–4 |
| PAA7 | 3.45 (0.67) | 3.67 (0.53) | 3.39 (0.66) | 3.68 (0.53) | 3.35 (0.59) | 3.35 (0.62) | 3.15 (0.81) | 3.42 (0.81) | 19.9 | <0.01* | 4.54 | <0.05* | 4.87 | <0.05* | 1–4 |
| PAA11 | 2.59 (0.77) | 2.76 (0.80) | 3 (0.80) | 2.72 (0.83) | 2.59 (0.81) | 2.71 (0.84) | 2.40 (0.82) | 2.60 (0.78) | 2.27 | 0.13 | 1.13 | 0.33 | 1.33 | 0.25 | 1–4 |
| Factor 2 | | | | | | | | | | | | | | | |
| PAA13 | 1.73 (0.63) | 1.72 (0.57) | 1.96 (1.02) | 1.69 (0.74) | 1.78 (0.60) | 1.87 (0.63) | 1.95 (0.61) | 1.90 (0.64) | 0.09 | 0.75 | 2.96 | <0.05* | 2.25 | 0.06 | 1–4 |
| PAA12 | 2.23 (0.69) | 2.41 (0.82) | 2.52 (0.79) | 2.23 (0.83) | 2.19 (0.68) | 2.34 (0.74) | 1.95 (0.69) | 2.38 (0.58) | 3.31 | 0.06 | 0.51 | 0.67 | 1.37 | 0.24 | 1–4 |
| PAA8 | 1.83 (0.70) | 1.74 (0.68) | 2 (0.90) | 1.62 (0.72) | 1.89 (0.50) | 1.82 (0.65) | 2.05 (0.69) | 1.85 (0.58) | 6.38 | <0.01* | 2.65 | <0.05* | 3.41 | <0.05* | 1–4 |
| PAA10 | 1.91 (0.88) | 1.87 (0.72) | 2.26 (0.81) | 1.84 (0.79) | 2.19 (0.70) | 1.97 (0.74) | 2.15 (0.59) | 2.08 (0.72) | 5 | <0.05* | 2.41 | 0.06 | 3.22 | <0.05* | 1–4 |

Notes:
* Statistically significant.
F, female; M, male; SD, standard deviation; F, Fisher test value; P, p-value.

inadequate solutions to promote PA during the lockdown, and adolescents did an insufficient amount of PA.

## DISCUSSION

The main purpose of the present study was to evaluate the effects of social restrictions on citizens' habits after the COVID-19 quarantine and to investigate its rationale.

As predicted, we found that the prevalence of individuals who did PA decreased after lockdown, which should indicate a negative effect of quarantine on people's habits.

The psychological aspects were the most common problem that affected the Italian citizens.

Generally, we observed that the PA prevalence decreased after the lockdown, which is in agreement with some authors who found that the lockdown negatively affected vigorous and moderate PA measured as METs (metabolic equivalents) per week (*Hargreaves et al., 2021*). As regards gender and age, the PA prevalence has fallen after the lockdown in both male and female subjects and in citizens who were 40 or more, but only the number of active women has decreased from lockdown up to the period after quarantine. Although the beginning larger number of women could have influenced the observed gender discrepancy, some authors suggested that this significant female decline could reflect the fact that women were more negatively affected by COVID-19 fear-related, which increased depression, anxiety, and stress levels and impacted their lifestyle habits (*Koçak, Koçak & Younis, 2021*). Our results showed that female citizens experienced many psychological problems than males, which could be the most relevant rationale that has influenced their PA habits. However, although several studies have investigated people's habits' differences during the lockdown, to the best of our knowledge few researchers analysed whether the citizens' PA practice varied after the quarantine, and no one analysed how the prevalence of PA varied. Of these, just one study showed a female decreasing trend in PA prevalence, but the authors examined the habits of Italian older adults (≥65 years) and considered the PA responses as levels increment (*Gallè et al., 2021*). Differently, our results are comparable with many studies that considered the differences between the period before and during the lockdown. Of these, one study suggested that most of the sample (679 Italian adults) maintained their training habits during the lockdown, and people who remained active tend to experience higher levels of energy and calm, and lower levels of perceived fatigue (*Di Corrado et al., 2020*). In addition, one research did not show discrepancies in PA prevalence and suggested that both men and women tried to stay active in quarantine (*Mauro et al., 2022*). However, we observed that the reduction in PA prevalence during the lockdown was more evident in older adults, in agreement with the above-mentioned results.

Concerning age, just one longitudinal study showed that both younger and elder adults were more active (30 min to 2 h of training per week) after the home forced period (*Bu et al., 2021*). Despite our study focusing on the PA prevalence and did not investigate its amount, our results suggested that the number of active individuals younger than 49 decreased after lockdown, while the number of older people had a larger decrement during the lockdown. These results could suggest that younger individuals have been most negatively affected by long-term effects related to quarantine. In addition, we found that many youths selected the socio-economics problems as motivation that conducted them to not practice PA after the quarantine. Differently, several studies agreed with our outcomes and found that the PA levels decreased in elder people during the lockdown, which may represent a relevant threat to public health (*Giustino et al., 2020*; *Ferrante et al., 2020*; *Gallè et al., 2021*; *Mauro et al., 2022*). Although the psychological aspects (meant as problems and/or benefits) were the main reasons that influenced the PA practice, many elders motivated their inactivity through health problems, which could indicate they saw the PA

practice as harmful to their disease status. Differently, other authors showed a PA reduction in university students and an increased usage of sedentary behaviours (*Gallè et al., 2020a*, *2020b*). Although the practice of daily PA should be promoted to any individual, our data suggest that great attention should be focused on promoting healthy habits in people who were elder than 40. These categories are more sensitive to cardiovascular, metabolic and cancer diseases. Also, the 40–50 years classes could be considered the target sensible age range in which began the senescent decline of the appendicular muscle mass and functional strength, which could cause Sarcopenia, obesity, fragility, muscle-skeletal injuries, hospitalization and decline in quality of life (*Cruz-Jentoft et al., 2019*). In addition, a recent review showed that quarantine had the most negative impact on lifestyle habits in people who were affected by pathologies such as diabetes, obesity, Parkinson's, and cardiovascular and neuromuscular diseases (*Zaccagni, Toselli & Barbieri, 2021*).

The second aim of this study was to understand what people thought of physical inactivity's effects on public health and whether they retained the restrictive measures enforced by the Italian government were adequate to favourite the PA habits.

We found that younger and female people considered that the lack of PA and/or sports practice negatively impacted the psycho-emotional dimension during the lockdown, especially in children/adolescents (items 6 and 7). Differently, both younger and older adults retained that the lack of PA and/or sports practice negatively affected the psycho-emotional status of elder individuals, but women resulted more sensitive than men (item 9). Several studies showed that quarantine led to deteriorated psychological aspects such as stress, anxiety, depressive symptoms, social isolation, and mood (*Becerra-García et al., 2020*; *Antunes et al., 2020*; *Stanton et al., 2020*). Consistent with our outcomes, one of these found that both women and people who were younger than 34 revealed higher scores for the anxiety state and trait (*Antunes et al., 2020*). In addition, Stanton and collaborators (*Stanton et al., 2020*) found that women, young adults, and people with chronic illnesses showed higher levels of anxiety, stress, and depression, and higher psychological distress was associated with negative lifestyle changes. Recently, some authors suggested that younger adults may spend a lot of time thinking and worrying about the outbreak, which may lead to general anxiety disorders and mental illness (*Huang & Zhao, 2020*). Also, one recent research showed that younger adults could be distressed by quarantine secondary consequences such as the social standstill and the economic decline, because they attempted to tackle many of life's key transitions (educational, professional, social, *etc.*), but they were temporarily frustrated in these efforts (*Shanahan et al., 2020*). As regards the gender differences, some researchers found that female psychological distress was higher than men, and female athletes reported higher levels of depression, loss of energy and demotivation during COVID-19 lockdown (*Pillay et al., 2020*; *Shechter et al., 2020*). These gender disparities in psychological and emotional distresses could be due to genetic and epigenetic factors (*McLean & Anderson, 2009*). However, our results emphasized that the youngest citizens were more worried than elder individuals about public health and the consequences of unhealthy habits due to physical inactivity (items 2 and 3). In addition,

younger women were more concerned about consequences caused by the lack of PA and/or sports practice in the national health system (item 5).

As regards the government measures adopted to slow the virus spread, we found that younger and female citizens disagreed with socio-political decisions more than male and older individuals, and they retained that the Italian government did not adequately consider the role and importance of PA and/or sports practice (item 10). Also, younger citizens retained that the adolescents did not practice enough amount of PA and/or sports during the lockdown (item 8), and they believed that school PA promotion was inadequate to involve them in exercise (item 13). Although many studies investigated on lives' habits during the lockdown, to the best of our knowledge no similar results were published. However, governments should make more effort to promote and enhance healthy habits and improve public health after the pandemic status due to COVID-19. Several strategies were proposed by previous studies to promote the PA practice through policies, schools, hospitals, transport, and the public sector, accessibility to adequate places, communications campaigns, and social support (*National Academy of Sciences, 2015*; *Tuso, 2015*; *Roundtable on Obesity, Food and Nutrition Board, Institute of Medicine, 2015*; *CDC, 2022*).

This study presents many limitations as the fact that we did not analyse how the amount of PA varied, and we could not exclude the presence of other motivations not correlated with quarantine, which may have affected the PA prevalence. Also, a larger sample size could have explained more evidence and the unbalanced gender size may affect the robustness of the questionnaire. In addition, we did not request the individual medical record, and we did not know whether each participant was affected by the Sars-Cov-2 during or after lockdown. Finally, we did not assess the test-retest reliability in the whole sample, but just 149 citizens were tested twice. Further studies that may fill these gaps are needed.

## CONCLUSIONS

Italian citizens lived at least 3 months of home-forced quarantine, which could have had negative consequences on their lifestyle habits. The physical activity prevalence reduced after the lockdown, and this may represent any trouble for public health. Despite this, citizens who were active during the lockdown experienced both psychological and physical well-being related to the practice of physical activity. So, promoting healthy behaviours as the practice of physical activity could be a cheap and easy solution to face negative quarantine consequences on public health. The government's attention should focus on adequate preventive strategies to improve citizens' health.

### Funding
The authors received no funding for this work.
## Competing Interests

Gianpiero Greco is an Academic Editor for PeerJ.

## Author Contributions

- Mario Mauro conceived and designed the experiments, performed the experiments, analyzed the data, prepared figures and/or tables, and approved the final draft.
- Stefania Toselli conceived and designed the experiments, authored or reviewed drafts of the article, and approved the final draft.
- Silvia Bonazzi performed the experiments, prepared figures and/or tables, and approved the final draft.
- Alessia Grigoletto performed the experiments, prepared figures and/or tables, and approved the final draft.
- Stefania Cataldi analyzed the data, authored or reviewed drafts of the article, and approved the final draft.
- Gianpiero Greco analyzed the data, authored or reviewed drafts of the article, and approved the final draft.
- Pasqualino Maietta Latessa conceived and designed the experiments, prepared figures and/or tables, and approved the final draft.

## Ethics

The following information was supplied relating to ethical approvals (*i.e.*, approving body and any reference numbers):

The study was approved by the Bioethics Committee of the University of Bologna (Approval code: 224254, October 9, 2020).

## Data Availability

The raw data are available as a Supplemental File.

## Supplemental Information

Supplemental information for this article can be found online at http://dx.doi.org/10.7717/peerj.14123#supplemental-information.

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
