# Peer review of "Effects of quarantine on Physical Activity prevalence in Italian Adults: a pilot study"

_PeerJ, doi:10.7717/peerj.14123_

## Round 0.1 · original submission · Minor Revisions

The reviewers agreed on the merit of the article. However, some changes must be undertaken to ensure better clarity and final quality.

·

Basic reporting

Dear authors;
First of all, I would like to thank you for reviewing this article. Also, after having carried out the review, I believe that it shows very relevant data on a time that is really worrying from a social, physical and psychological point of view. In terms of the design and theoretical basis of the research, it meets the quality criteria, with the data collection instruments showing a high degree of reliability. However, I believe that some minor modifications should be made, which I will mention below:
Line 28: Add the male sex as well. I also suggest adding the number of participants of each sex along with the %.
Line 30: Replace Elder with older.
Material and methods section, specifically in Participants and Study Design, add the distribution by sex.

Experimental design

I consider that the experimental design is well justified and developed in the manuscript, so no modification or clarification is necessary.

Validity of the findings

The instruments used show a high degree of reliability. The data also meet the research objective.

Additional comments

I consider the changes to be minimal. Good empirically and theoretically well-developed research is presented.

Reviewer 2 ·

Basic reporting

You have written an interesting paper. In general, the manuscript is clearly written. The literature used is current and well in line with the research topic.
The introduction leads well to the main rationale of the paper.
The structure of the paper is ok.
Paper needs English profreading

Experimental design

The design of the paper is in line with the PeerJ journal.

The main aims of the study are well established and clear and highlight the importance of developing a new questionnaire.

Material and methods

How was your sample size determined (G*Power or any other method)? Please report

Line 99 - Why do you state that written informed consent from parents was obtained if you stated that participants were 18 and older? Please elaborate

How long did, on average last to fill the questionnaire? Add info

How many questionnaires were completely filled and how many had missing questions? Please add info

Line 125-126 - add references on which you decided on the used age groups. Also, from table 1 or from the results, the number of participants for each age group is not clear. Please add this info in table 1

Line 143 - add a reference for risk factor and briefly describe how it was calculated

You mention in the methods that you tested test-retest reliability. On what sample of participants, what was the time difference between questionnaires? Please report

The discussion is well rounded up, addresses all relevant results, and compares them to relevant literature.

However, the limitations section should also address the unbalanced sample as from the point of gender and from the point of age groups. This could add to the robustness of the questionnaire and should be addressed in the future development of questionnaires of this kind.

Validity of the findings

Statistical analysis is sound and the main rationale and aims were achieved.

The discussion is well-rounded with the necessary literature.

Conclusion is well stated and is supported by the resučts of the study.

Additional comments

The manuscript is well written and is going in the right direction. However, some small parts still need to be addressed for greater clarity. Therefore some minor additional revision is needed.

Kind regards

---

## Round 0.2 · accepted · Accept

I can confirm that all the comments were addressed in the current revised document and that the article holds the necessary merit to be published.